# Digital-Based Policy and Health Promotion Policy in Japan, the Republic of Korea, Singapore, and Thailand: A Scoping Review of Policy Paths to Healthy Aging

**DOI:** 10.3390/ijerph192416995

**Published:** 2022-12-17

**Authors:** Nadila Mulati, Myo Nyein Aung, Malcolm Field, Eun Woo Nam, Carol Ma Hok Ka, Saiyud Moolphate, Hocheol Lee, Yuki Goto, Nam Hae Kweun, Takumi Suda, Yuka Koyanagi, Yuiko Nagamine, Motoyuki Yuasa

**Affiliations:** 1Department of Global Health Research, Graduate School of Medicine, Juntendo University, Hongo 2-1-1, Bunkyo Ku, Tokyo 113-8421, Japan; 2Advanced Research Institute for Health Sciences, Juntendo University, Hongo 2-1-1, Bunkyo Ku, Tokyo 113-8421, Japan; 3Faculty of International Liberal Arts, Juntendo University, Tokyo 113-8421, Japan; 4Faculty of Social Sciences, Kyorin University, Tokyo 181-8611, Japan; 5Faculty of International Liberal Arts, Waseda University, Tokyo 169-0051, Japan; 6Department of Health Administration, Software Digital Healthcare Convergence College, Yonsei University, Wonju 26493, Republic of Korea; 7Gerontology Programmes & Senior Fellow (Service-Learning & Community Engagement), Centre for Experiential Learning, S R Nathan School of Human Development, Singapore University of Social Science, 463 Clementi Road, Singapore 599494, Singapore; 8Department of Public Health, Faculty of Science and Technology, Chiang Mai Rajabhat University, Chiang Mai 50300, Thailand; 9Department of Global Health Promotion, Tokyo Dental and Medical University, 1-5-45 Yushima, Bunkyo-ku, Tokyo 113-8510, Japan; 10Department of Family Medicine, Graduate School of Medical and Dental Sciences, Tokyo Dental and Medical University, 1-5-45 Yushima, Bunkyo-ku, Tokyo 113-8510, Japan; 11Department of Preventive Medicine, Wonju College of Medicine, Yonsei University, Wonju 26426, Republic of Korea; 12Research Team for Social Participation and Community Health, Tokyo Metropolitan Institute of Gerontology, 35-2 Sakae-cho, Itabashi-ku, Tokyo 173-0015, Japan; 13Department of Judo Therapy, Faculty of Health Sciences, Tokyo Ariake University of Medical and Health Sciences, Tokyo 135-0063, Japan; 14Division of the Health for the Elderly, Ministry of Health, Labour and Welfare, 1-2-2 Kasumigaseki Chiyoda-ku, Tokyo 100-8916, Japan

**Keywords:** digitization, aging, global health, long-term care, social welfare, policy, Asia, gerontology, DIHAC

## Abstract

People are living longer, and our life has become more digital. Hence, the benefits from digital technology, including economic growth, increasing labor productivity, and ensuring health equity in the face of an aging population emerged as a vital topic for countries around the world. Japan, the Republic of Korea (ROK), Singapore, and Thailand are in the top ten rankings in terms of information and communication technology (ICT) development within the Asia Pacific Region and all are facing challenges of population aging. Well-designed national ICT policy and health promotion policies enabled the countries to make significant progress and development in terms of digitalization and healthy aging. This paper aims to answer questions regarding digitization and health promotion: when it started, how it is going, what are the achievements, and what it holds for the future, considering healthy aging and digitalization by reviewing the national ICT policy and health promotion policies of Japan, Korea, Singapore, and Thailand. This paper is expected to help readers build a comprehensive understanding of each country’s journey towards building a healthy aging digital society. Furthermore, we hope this paper can be a source for countries to exchange experiences and learn from each other with a joint goal of building a healthy aging digital society.

## 1. Introduction

Population aging and digitalization are the two main trends happening in the world. The share of the global population aged 65 years or over is expected to increase from 10% in 2022 to 16.0% by 2050 [1]. Meanwhile, there were an estimated 4.9 billion (63%) people worldwide using the internet in 2021, compared to 4.1 billion internet users in 2019 [2]. Exacerbating the need has been the COVID-19 pandemic, which has accelerated the integration of digital technology into our everyday life as most activities were moved online. This highlighted the necessity for older adults to adopt and embrace digital technology [3] and is one of the pathways toward healthy aging in this digital era [4].

Japan, Singapore, Korea, and Thailand are the four countries that are in the top ten ranking in terms of Information and Communication Technology (ICT) development within the Asia Pacific Region [5], and all are facing challenges of population aging. As the “super-aged society”, the proportion of older adults in Japan constituted 28.8% of the total population in 2020 [6]. In 2025, Korea will become a “super-aged society” as well, as its aged population increases from 17.5% in 2021 to 20.6% by 2025 [7]. Singapore is an aged society, with 16% of its population aged 65 years and over in 2021 [8], and this proportion is expected to increase in the near future. Thailand is one of the fastest-aging countries, where 12.6% of the total population were aged 65 years and over in 2021 [9]. Technologically, the internet usage of individuals in these four countries also exceeded the world average; 97% in Korea, 92% in Singapore, 83.4% in Japan, and 86.6% in Thailand. However, in all four countries, older adults consist of a small proportion compared to the younger age group population in the digital world [10,11,12,13].

Reducing the digital divide among older adults has become one of the factors to achieve healthy aging, and it also contributes to fostering health equity among older adults [14]. A well-designed national policy is a path to building a society where every older adult is healthy and digitally connected. There is great number of papers focused on the digital-based policy or health policy of these four countries individually or collectively. However, few have discussed the two policies together with the goal to highlight the necessity of closing the digital divide among older people to achieve healthy aging in this digital era. Therefore, this policy review paper is part of the ongoing cross-cultural study, “Digitally inclusive, healthy aging communities (DIHAC): a cross-cultural study in Japan, Singapore, the Republic of Korea, and Thailand” [15], which aims to review and summarize the development of the ICT-based policy and national health promotion policy of the four countries. Through this paper, we aimed to discover the following about the four countries: the beginning of the digitization and health promotion journey, its achievements, its current status, and its future direction in terms of the digitally inclusive, healthy aging community.

This paper is divided into two main sections. The first section mainly focuses on the digitization journey, and the second section focuses on the health promotion policy of Japan, Korea, Singapore, and Thailand. It is important to note that we are not trying to compare the four countries, but to review each to build a comprehensive understanding of the countries’ journeys and current status upon which to base the DIHAC study [15].

## 2. Methods

### 2.1. Research Question

The focus of this research was the digitalization and health promotion journey of Japan, Korea, Singapore, and Thailand through the lens of the past, present, and future answering such questions as: When did it start? How it is going? What are the achievements? Finally, what future direction will be considered with the fast-aging population and fast pace digitization?

### 2.2. Data Source and Search Strategy

The review for this project comprised of a literature review of secondary sources focusing on ICT-based national policies and national health promotion policies of Japan, Singapore, Korea, and Thailand. The literature review has included national policy and planning documents, academic and research articles, annual reports, and published papers.

The search strategy is mainly focused on official publications from government websites regarding ICT and health promotion. The search process started with visiting the official websites first and looking for the publication, then selecting the published papers or reports according to the keywords (digitization, ICT, policy, health, health promotion, development, annual report, white paper, statistic). A small number of the sources were found via search engine, Google Scholar; the keywords for searching were: (1) country name ICT, (2) country name digital, (3) country name ICT policy, (4) country name digital policy, (5) country name health promotion, and (6) country name health promotion policy.

The main database for each country is summarized in the Table 1 below.

In addition, a regular policy review meeting was held internationally via Zoom, among the research teams from four countries monthly from April 2021 to December 2021 and bimonthly in 2022. Interactive discussions were noted as summaries to reflect in this scoping review [16]. The study has been ethically approved by Juntendo Medical Ethics Committee. The approval number is E22-0057-M01.

### 2.3. Eligibility Criteria

The reports, white papers published by the government, and official entities regarding the policy of ICT development and health promotion were included in the review process. Other papers that aimed to discuss the overall ICT or health promotion policy, its development, and current status were included to review in this paper. The search was not conducted in a systematic manner, instead it focused on the broad topic of overall path towards digitization and health promotion.

## 3. National Digitization Journey

### 3.1. The Past

Japan: In 1958, the first computer was introduced to the government of Japan [17], and in the following decade, the Japanese government strived to increase the effectiveness of the public system through the use of computer technologies [17]. The digitization of Japanese society started in 2000, when the first national law “IT Basic Act” was enacted, followed by the introduction of the first national strategy, “e-Japan strategy”, a five-year national strategy that aimed to become the world’s leading IT nation by developing the ICT infrastructures [18]. With the proposal of the “New IT reform strategy” in 2006 to build a society where ICT is accessible to anyone anywhere anytime [18], followed by the proposition of the “i-Japan Strategy 2015” in 2009, Japan promoted the utilization of ICT. The next stage focused on the utilization of digital data, starting in 2013. Although Japan was slow to utilize the capacities of the ICT revolution, it declared it was to become the world’s most advanced IT nation and also formulated the “Digital Government Strategy” in 2017 to further foster the digitalization of public administration. The local governments and private sectors also embraced information technology more through the “Digital Government Action Plan” in the following year. Despite the effort, the outbreak of the COVID-19 global pandemic highlighted Japan’s slow pace of digitization. Therefore, in 2021, “the Basic Act on Forming a Digital Society” was enacted, along with the formation of a Digital Agency to build a digital society [19,20].

Korea: The use of the computer to minimize manual work, and increase the efficiency of the public task can be traced back to 1967 when Korea introduced the computer in its national census [21], which is a starting point of the computerization of public administration. This was followed by the announcement of the Master Plan for the Computerization of Administration (1975–1982) by the Ministry of Government Administration, the start of the nation’s First Five-Year Master Plan for the Computerization of Administration (1978–1982), and in June 1979, the government issued the Guidelines for the Computerization of Administration. This provided the legal basis for the implementation of the plan with the steps taken by the country towards more efficient administration of public tasks in every sector [21]. To build a national information and communication network, the Act on Expansion of Dissemination and Promotion of Utilization of Information System was enacted in 1986, and the National Basic Information System (NBIS) Project was started in the following year. The NBIS project consisted of five sub-systems, divided into two stages, stage one: 1987~1991, and stage two 1992~1996, symbolizing the beginning of the nation’s journey to informatization [21,22]. The Ministry of Information and Communication (MIC) was the main government agency to promote digitization from 1994 to 2008, and contributed greatly to the digitization of the country by making frameworks and policies, such as the Framework Act on Informatization Promotion, Comprehensive Implementation Plan for Building a High-speed Information and Communication Infrastructure [21]. In 2002, Korea started e-government promotion, strengthened the efficiency of public administrations and civil participation, and increased economic feasibility [23]. As one of the leading countries in ICT development and e-Government, the government of Korea established clear visions and strategies over the years. The first informatization Promotion Master Plan (1996–2000), the second Cyber Korea 21 (1999–2002), the third e-Korea Vision 2006 (2002–2006), the fourth National Informatization Master Plan (2008–2012), and the fifth National Informatization Master Plan (2013–2017) were mid/long-term national informatization master plans aimed to build a digital society [24] and layered the foundation to the sixth National Informatization Master Plan (2018–2022) for an Intelligent Information Society [25]. The COVID-19 pandemic has highlighted the rapid development of Korea’s digitization. Although the systems have had some ‘hiccups’, Korea was able to rapidly use digital technology to share related information to the public, contact tracing systems, mHealth, e-education, telework, etc.

Singapore: The “National Computerization Plan” of Singapore started in 1980. As the first national master plan, it aimed to develop the IT industry and labor, and increase the efficiency of the government departments and public sectors’ services by using computer technologies [26]. After the successful completion of the first master plan, the second national master plan, the “National IT plan” was formulated from 1986 to 1991. This plan focused on the digitization of private sectors, local companies, and IT industries intending to further develop the IT industry and facilitate the adaptation of digital technology [27]. The third national master plan, the “IT 2000”, was proposed from 1992 to 1999, and it envisioned transforming Singapore into an intelligent island, where information technology was integrated into every aspect of Singaporean life, increasing the quality of life of its citizens, boosting economics, becoming a global hub for business, transportation, and services, linking local and global communities, and enhancing individual potential [27]. The fourth national plan, known as “Infocomm 21” (Information and Communications Technology for the 21st Century) ran from 2000 to 2003 and aimed to build a digital society holistically by developing an e-Economy, e-Society, and new growth engine. Infocomm Development Authority of Singapore (IDA) was formed as the main agency to focus on the strategy and policy of the development of information and communication technology in Singapore [28]. With a vision to turn Singapore into the Intelligent Nation and Global City, a ten-year national master plan, the “Intelligent Nation 2015” was proposed in 2006 [29]. Four key strategies were included to not only boost the nation’s economic competitiveness, but also to close the digital divide, and let every citizen benefit from ICT in their daily lives [29]. In 2014, Singapore launched the “Smart Nation Initiative” to transform five domains of its nation (health, education, transport, urban solutions, and finance), by putting effort into three pillars (digital society, digital economy, and digital government), with the contribution of public, private, and people sectors [30]. This joint effort resulted in Singapore becoming the world’s ‘smartest’ city over three consecutive years from 2019 to 2021. Along with the above-mentioned national master plans, from 1980 to 2011, the five e-Government master plans, “Civil Service Computerization Plan”, “e-Government Action Plan 1&2”, “iGov 2010”, and “eGov 2015” were proposed to support the national ICT master plans, and support the efficacy, efficiency, integration, and innovation of the government [31].

Thailand: The digitization of Thailand can be traced back to the proposal of the first national IT policy (IT 2000) in February 1996. This five-year policy framework laid the foundation for Thailand to adopt ICTs into its administrative and business practices. The objective was to develop the ICT infrastructure of Thailand, integrate information technology into public sectors, promote the growth of local industries, and build human capacity, by increasing the digital literacy of its population [32]. The second national policy framework “IT 2010”, started from 2001 to 2010, encompassing three key components: building knowledge-based human capital; promoting innovation in economic and social systems, and strengthening information infrastructure and industry to build a knowledge-based society. Five key strategies focused on e-Government, e-Industry, e-Commerce, e-Education, and e-Society were formulated [33]. In 2002, The Thailand National Master Plan (2002–2006) started, and it outlined four main goals and seven key strategies as a measure of the achievements of IT 2010 [32]. Considering the development state internationally and domestically, Thailand envisioned becoming a smart nation and launched the third national framework, the “IT 2020”, a ten-year plan in 2011 [32].

The current national ICT-based policy of the four countries is summarized in Table 2.

### 3.2. The Present

Well-designed national digital policies throughout the years led Japan, Singapore, Korea, and Thailand to achieve much progress in the implementation and uptake of digital technologies. Throughout the implementation, the ‘hardware’ delivery of the digital revolution also changed, becoming more ubiquitous with the arrival of smaller and more powerful personal mobile devices.

#### 3.2.1. Infrastructure and Access

In Japan, Singapore, Korea, and Thailand, ICT infrastructure and access to mobile cellular subscriptions are far over 100 per 100 inhabitants. Active mobile broadband subscriptions per 100 inhabitants are estimated by International Telecommunication Union (ITU) to have reached 203 per 100 inhabitants in Japan, 155.6 per 100 inhabitants in Singapore, 114.9 per 100 inhabitants in Korea, and 86.7 per 100 inhabitants in Thailand. Korea, Japan, and Singapore have all attained fixed broadband subscription rates higher than the global average (15.2%). Thailand achieved a compound annual growth rate (CAGR) above 10% between 2015 and 2019 (12.5%) [34].

#### 3.2.2. Internet Use

According to International Telecommunication Union, the percentage of internet users in Korea, Japan, Singapore, and Thailand exceeded the world average (63%), with a proportion of 97%, 92%, 90%, and 78%, respectively [35]. Moreover, the internet penetration rate ranges from nearly 100% in Korea (99.7%), Singapore (98.7%), Japan (96.6%), to nearly 75% in Thailand (74.6%), all above the world average penetration rate [34].

#### 3.2.3. Enabler or Barrier to Internet Use-Digital Skill

Digital skills can be a facilitator or a barrier for an individual to adopt digital technology. According to the “ITU Digital trends in Asia and the Pacific 2021” report, Korea is at the top by far in Asia and the Pacific Region for both basic (73.1%) and standard (50.5%) ICT skills, followed by Japan (59.9% basic, 48.8% standard ICT skill), Singapore rated 4th (52.7% basic, 41.9% standard ICT skill), and Thailand rated 11th (15.6% basic, 9.8% standard ICT skill) [34].

#### 3.2.4. E-Government Development

To consider the development of the e-Government, and the sustainability, inclusiveness, and equitability of the government public service we have utilized The United Nations e-Government Development Index (EGDI), comprising assessment of the Telecommunications Infrastructure Index (TII), Human Capital Index (HCI), and Online Service Index (OSI) [36]. According to this index, Korea (2nd), Singapore (11th), and Japan (14th) lead in e-government development in the world. Thailand, although ranked 57th, joined the very high EGDI group for the first time in 2020. These four countries are in the top 15 countries in terms of e-Government performance [36]. These achievements show the evidence-based effectiveness of the strategic policies developed by each country over the years.

### 3.3. The Future

Notwithstanding statistics, the new digital era requires every individual, regardless of age or socioeconomic status, to be connected to digital technologies. This must also be achieved if we are to realize the United Nations Sustainable Development Goals of closing the digital divide, leaving no one behind. However, the “gray digital divide”, a digital gap between the younger and older generation exists in every country. In Japan, more than 90% of individuals aged 13–59 years use the internet, whereas, for people in the older age groups, this proportion remains relatively small, diminishing as age increases: age 60–69 (82.7%), age 70–79 (59.6%), and age 80 and over (25.6%) [37]. In Singapore, individual connectivity is 100% for those aged 15–24, 94% for those aged 25–74, and 90% for those aged less than 15. Although the youth made up the biggest portion of internet use in 2020, only 46% of individuals age 75 and over used the internet in 2020 [11]. In Korea, similar statistics are evidenced, with more than 95% of individuals aged 10–59 years, and 82.5% of adults in their 60s being internet users; however, only 31.8% of adults aged 70 years and over used the internet [10]. In Thailand, people aged 15–24 years used the internet the most at 98.8%, followed by those aged 25–39 years at 98.1%, while people aged 60 years and over had the lowest Internet use at 52.5% [13].

As Japan, Singapore, Korea, and Thailand are all facing the demographic shift of the aging population, to prevent isolation from society, maintain productivity, and ensure the healthy aging of its older adults, policies and programs to close the gray digital divide, using ICT to benefit all parts of society are necessary. These four countries have recognized the urgency and necessity for digital uptake and literacy in their aging societies, and they have developed national programs to promote the adaptation of digital technology among their senior citizens.

By way of example, Singapore’s “Seniors Go Digital” national program assists older adults to learn to use basic communication tools for messaging and to make video calls, learn to access government digital services, learn to use e-Payment tools and internet banking apps, and learn cybersecurity tips [38]. “Infocomm Media Development Authority (IMDA) Digital Access Programme” is aimed to close the first-degree divide access divide by providing mobile access for seniors who wish to learn the digital skills but lack financial means to do so [39]. There is also the “Intergen IT Bootcamp” that encourages the intergenerational learning of ICT skills, and “Silver Digital Creators (SDC)” is for seniors who have some basic digital knowledge and want to upgrade their skills and learn creative digital skills. “Silver Infocomm Junctions” are learning hubs located island-wide for seniors to start their digital learning journey. The “Tech Connect” initiative aimed to help older adults be Smart Nation ready [40].

In Japan, there are two types of digital skill training, national expansion type and regional cooperation type for older citizens. Mobile phone carrier companies (Docomo, au, Softbank, etc.) are the main entities, and shops around the country provide support for older adults to learn how to use mobile phones for administrative procedures and government services. The regional cooperation type is mainly with the cooperation of local government and located mostly in city halls, community centers, and silver citizen clubs to provide support on basic digital skill learning, along with e-government services [41,42]. *Routeku* (Geriatric Technology) Research Group of NPO Broadband School Association has been working to help community-dwelling older residents learn digital technology for over 25 years [43]. Volunteers, mostly older people from the association use easy, fun methods to introduce digital technology to their peers. Intergenerational interaction activities were also organized in different communities to learn about programming [44]. Digital *Hinamatsuri* (also known as girl’s day, to celebrate and pray for the health and prosperity of girls every March 3rd) has been organized since 1997. During the “stay-at-home” measure taken to prevent the spread of COVID-19 in 2020, NPO Broadband School Association organized online seminars for creating a mutual help system for natural disasters.

In the Korea, the National Information Society Agency Department of Digital Inclusion is the main body to implement policies and programs for a digitally inclusive society. “The Digital Inclusion Act” plans to build an inclusive society [45]. Currently, digital education for older adults and other vulnerable groups is carried out in welfare centers, district offices, and community service centers. The main education content is basic digital skills used in everyday life, cyber security, etc. [45]. Personal computers have also been distributed to those who are in need. Private technology companies are also collaborating by designing age-friendly digital devices and special services for older adults. The Korea Information Society Agency developed “50+ site” and “National Informatization Education site” for digital capacity building for its older citizens [46,47].

In Thailand, to help older adults adopt digital technology, the Electronic Transactions Development Agency (ETDA) provides a “digital citizen course” for Thai seniors. The agency not only produces educational materials, videos, and tool kits, and organizes digital education programs, but also seeks to train individuals to get trainer certificates, which enable them to continue spreading digital knowledge in their community [48]. Some schools are offering computer and social media courses to older people [49]. To speed up the process of building a digitally inclusive society, the “Digital Community Center (DCC)” project of Thailand aimed to build a learning center for all citizens in communities across all regions of the country, with 2277 centers located in various rural areas [50].

## 4. National Health Promotion

The health state of an individual is determined by multiple factors, the social and economic environment, the physical environment, and the characteristics and behavior of the individual [51]. Governments play an even more crucial role in the well-being of their population by making policies at the state or national level, strengthening governance that supports affordable and accessible healthy choices for all [52], and guiding the nation’s population to gain control over their health.

In the previous section, we focused on the digital policy of Japan, Korea, Singapore, and Thailand. In this section, we are going to focus on the other pillar of this paper, which is the development of the national health promotion policies of these four countries. The life we are living is becoming more and more digital, and integration of digital technology into health promotion activities might become a must to make sure of its sustainability and its accessibility, and realize healthy aging in the new digital era.

### 4.1. The Past

Japan: The Ministry of Health, Labour, and Welfare is the main government agency for health in Japan. In 1978, The “First National Health Promotion Measure” was released. This ten-year plan aimed to encourage lifetime health promotion by implementing health check-ups for all ages, establishing health promotion bases by establishing health centers, and increasing the manpower and dissemination and awareness of health [53,54]. The Second National Health Promotion Measure, the “Active 80 Health Plan” started in 1988, put special focus on the importance of physical exercise to improve the functional ability and independence of people aged 80 years or over, along with other lifestyle measures to promote the overall health of all age groups [55]. In 2000, the third national health promotion measure, the “National Health Promotion Movement in the 21st Century (Healthy Japan 21)”, started focusing on the prevention of lifestyle-related disease, the promotion of healthy lifestyles by setting specific targets, and developing indicators to measure the mid-term and long-term efficiency of the action, using public-private sector, and communities working together to achieve the goals [53]. At the same time, the long-term care insurance system introduced in Japan in 2000 enables older people to benefit from health, and welfare from different agents according to their own needs. Under this system, community-based long-term care prevention activities developed vastly. In 2020, 93% of prefectures and cities have community-based long-term care prevention activities for older adults. Physical exercise remains the main activity (56.1%), followed by hobby-based activities (16.5%), *Sawakai* (15.4%), which is a social gathering, and dementia prevention activities (4%) [56]. Moreover, the enactment of the “Health Promotion Act” in 2002 further promoted the goals for a healthy lifestyle and a healthy lifespan. Currently, the Fourth National Health Promotion Initiative is ongoing, called “Healthy Japan 21 (the second term)”, the goal of this ten-year plan that started in 2013 is to extend healthy life expectancy and reduce the health disparities of its citizens. Prefectural Health Promotion Plans were also developed to further promote the health of its residents [57].

Korea: The Ministry of Health and Welfare is the main agency responsible for the national health promotion activities in the Republic of Korea. In 1995, the “Health Promotion Act” was launched. It was the first national law that aimed to improve the health state of the population at local and municipal levels, and it focused on health education, nutrition improvement, early disease detection and prevention, and healthcare [58]. The National Health Promotion Fund was launched in 1996 to financially support national health promotion projects [59]. Based on the National Health Promotion Act, multiple National Health Promotion Plans (Health Plan) were developed: The Health Plan 2010 (2002~2010), the Health Plan 2020 (2011~2020), and the Health Plan 2030 (2021~2030). The health plan was reviewed every ten years and evaluated every fifth year, updated to be suitable for the existing context of the socioeconomic position and health of the people [60], to increase healthy life expectancy and improve the health equity of its people. Moreover, the notion of a Healthy City was introduced to the country in 1996; Healthy City Projects were being carried out to contribute to the health and well-being of the population [61]. Currently, Korea is in the first five-year phase of the third national health plan. This “Health Plan 2030 (2021~2030)”, has 28 target areas, 400 performance indicators to reach the goal of “health for all”, and outlines the health span of its citizens in a more detailed and precise way [60].

Singapore: The Ministry of Health is the main public bureau for managing the public health system in Singapore. In 1983, the sector formulated the “National Health Plan (10 years)”, which was aimed at helping the population become healthy, active, vigorous, and physically fit, by focusing on preventive medicine and health education [62]. In 1992, the first national healthy lifestyle campaign started. Its goals were to increase awareness of the different aspects of healthy living, impart enabling skills to the various segments of the population to practice healthy living, and create and foster a supportive social and physical environment to encourage healthy living among Singaporeans [63]. Since implementation, annual month-long campaigns have organized a range of activities, events, and awards to promote the goals. The program, which targets all age groups, also includes surveys and feedback to review the progress of its objectives [64]. In 2001, the Health Promotion Board (HPB) was established as the main people-centered sector responsible for developing programs, and guidelines for schools, workplaces, communities, and individuals, along with partnering with local and international organizations to promote the health and well-being of all populations [65]. The Healthy Living Master Plan Taskforce was formed in September 2012 to look into making healthy living accessible, natural, and effortless for all Singaporeans [66]. Other than that, there are a series of health and wellness programs for seniors organized by community partners, “Live Well, Age Well” is an example organized by the People’s Association and HPB to promote the health and well-being of its older citizens [67]. In March 2022, the Ministry of Health announced the “Healthier SG Plan” for its citizens. This plan focuses on the importance of primary health care and uses the life course approach to promote the well-being of the population targeting different age groups. There are five key features of the Healthier SG Plan, including “(a) mobilize family doctors to deliver preventive care for residents; (b) develop health plans that include lifestyle adjustments, regular health screening, and appropriate vaccination which doctor will discuss with residents; (c) activate community partners to support residents in leading healthier lifestyles; (d) launch a national enrolment exercise for residents to commit to seeing one family doctor and adopt health plan; (e) set up necessary enablers, such as IT, manpower development plan and financing policy to make Healthier SG work” [68].

Thailand: In 2001, the government of Thailand enacted the “Thai Health Promotion Foundation Act, B.E. 2544”, which is financed by a 2% tax on liquor and tobacco to support national health promotion activities [69]. Furthermore, the Thai Health Promotion Fund (ThaiHealth) was established as an autonomous organization to encourage and promote the health of the Thai population, and it was based on the national health policy. ThaiHealth is also responsible for the support and funding of projects that aim to promote the health of the people [69,70]. The enaction of the “National Health Act B.E 2550” (2007) is the remake of the institutional development of the nation to further accelerate the way to health for all [71]. The joint work of the Ministry of Public Health and ThaiHealth, along with the collaboration of public and private sectors and an already existing strong network of village health volunteers has resulted in the successful, sustainable integration of the health promotion projects at the national and community level [71].

The current national health promotion policies of Japan, Korea, Singapore, and Thailand are summarized in Table 3

### 4.2. The Present

It is almost a truism to state that increasing life expectancy benefits society and individuals more if the added years are spent in good health. “Health Japan 21 (the second term)” of Japan, “Health Plan 2030” of Korea, “Healthier SG” of Singapore, and “Twenty-year National Strategic Plan for Public Health” of Thailand are aimed at the healthier future of the nations. The continuous efforts of the countries in the area of national health promotion discussed above have resulted in Japan, Korea, Singapore, and Thailand an increase in both life expectancy and healthy life expectancy. According to the World Health Organization Observatory, life expectancy at birth in Japan is 84.26 years, in Korea is 83.3, in Singapore is 83.2, and in Thailand is 77 years old in 2019. Healthy life expectancy at birth for Japan, Korea, Singapore, and Thailand are 74.1 years, 73.1 years, 73.6 years, and 68.3 years, in 2019, respectively [72]. Compared to a decade ago, both life span and health span increased by more than three years. This indicates the success of the countries’ efforts toward the promotion of the health and well-being of older adults.

### 4.3. The Future

With the demographic shift of the aging population and the increasing necessity of ICTs in everyday life, countries have to formulate guidelines and policies to build a digitally inclusive healthy aging society. The COVID-19 pandemic highlighted the importance for older adults to adopt digital technology to stay connected to society. Moreover, making existing health promotion activities more sustainable might require the integration of ICTs.

In Japan, when the pandemic started, the “stay at home”, and “physical distancing” measures were taken to prevent the spread of the virus, hence the traditional community-based health promotion activities were forced to be paused. However, considering the potential harm to older adults’ well-being due to isolation, minimum social interaction, and the reduction of physical activity leading to frailty, digital technologies were being integrated into health promotion activities. In May 2020, the “National Center for Geriatrics and Gerontology Home Exercise Program 2020 (NCGG-HEPOP^®^)” were developed and introduced to older adults to keep them active and mentally well [73]. Exercise videos with explanations were uploaded to YouTube [74]. For those who lack internet access or skill, a home exercise booklet or DVD was distributed [73,74]. In some ways, we can consider this an extension of the morning *rajio taiso* (radio exercise) that is broadcast on public radio and television. During the pandemic, community-based social gathering activities organized in “salons” were also stopped. National Center for Geriatrics and Gerontology (NCGG) developed the smartphone application “Online Salone App” to provide activities to prevent social isolation and cognitive decline, and promote physical activity [73].

Korea is currently focusing on integrating telemedicine into the areas of dementia, depression, and loneliness among older people in preparation for the aging population. In 2017, Korea has announced the ‘National Responsibility for Dementia System’, in which the state is responsible for dementia. Korea has installed dementia care centers in 252 public health centers across the country and comprehensively manages all services such as 1:1 customized counseling, checkup, management, and services [75]. Currently, dementia care centers are providing dementia prevention education and treatment services to the elderly using digital therapy tailored to the digital age. Additionally, Korea launched a ‘remote medical assessment’ pilot for dementia patients in some areas, combined with a visiting nurse service [76].

In Singapore, the “Healthy Buddy” application was developed by SingHealth to help citizens manage their clinic visits, via the app’s Health Champ feature to track blood glucose, blood pressure, cholesterol, height, weight, and BMI. Users were able to receive customized advisories and health tips, get to know more about the specific condition through specialty care features, order medicines, and set reminders for them to take medicine or complete other health tasks [77]. The “Healthy 365” mobile application was developed by the Health Promotion Board of Singapore. Users can earn health points by scanning QR codes via the application, which can be used to purchase groceries, healthier meals, and drinks from partnered stores, by signing up for in-app challenges and health programs. Moreover, the application is compatible with fitness tracking devices to allow users to log their daily steps and active exercise time [78].

The Thailand Ministry of Public Health announced a five-year “eHealth Strategy” that is in line with Thailand 4.0; the 20-year national strategy, National Economic and Social Development Plan aims at improving the quality of life for all Thai citizens by developing strong, equal, and efficient eHealth strategies by 2020 [79].

## 5. Result and Discussion

Japan, Korea, Singapore, and Thailand all play a significant role in the mapping of the global economy. With the cutting-edge innovativeness of Japan and Korea, the big and open bussiness hub of Singapore [80], the emerging economy of Thailand [81], and together being the leaders of ICT development in the Asia-Pacific region [5], the review and summary of the path they take in terms of digitization and health promotion is worth looking at.

### 5.1. National Digitization Journey

In the context of the fast-digitizing world, internet use has become a major social determinant of health [82]. Addressing any digital divide between the aged or aging populations with other aged groups in a society is essential to bring about the health equity of older adults, and perhaps reducing the knock-on burdens that may result from a non-healthy aging society. Take vaccine uptake for example, during the initial phase of COVID-19 vaccination in Japan, the vaccine rollout was impacted by the slow speed of registration. It needed digital skills and some older persons found it difficult to register by themselves. Help was required from friends, neighbors, or family members. This example showed how digital inclusion is important as a basic step in promoting access and equity of health services. Among the four ICT-related access gaps, material access, skill access, usage access, and mental access gap [83], Japan, Korea, Singapore, and Thailand addressed the material access gap the most. Network coverage has reached more than 95%, and internet access at home to more than 95% in Japan, Korea, and Singapore, although Thailand is still developing its infrastructure and is currently at 85% coverage. The digital skill, usage, and mental access among older adults needs more work, as they remain less connected through the internet compared to the younger generation. All four countries recognized the importance of national policies that are tailored to the reality of its current gray digital divide state to guide public and private sectors, and communities working together to help older adults become equipped with digital skills, knowledge, and confidence. Through this paper, it is hoped that we can provide a deeper understanding of the countries’ policies and how we can learn from each other in the evolution of the digital health world.

### 5.2. National Health Promotion Journey

Prolonged life span with good health benefits not only the individual, but also the community, society, and nation as a whole. The continuous effort towards the promotion of health in Japan, Korea, Singapore, and Thailand allowed its citizens not only to live longer but also live healthier. The use of digital technology to prevent frailty and loneliness in Japan during the COVID-19 pandemic, the telemedicine practices in Korea, the widespread use of mobile health (mHealth) in Singapore, and the promotion of eHealth in Thailand, all indicate the important role of digital technology in regards to promoting health and well-being of the population. In the context of health promotion, the integration of digital technology addresses the challenges of distance and cost, and the adoption of ICT by older adults amplifies the benefit. Along with implementing the policies and strategies towards closing the gray digital divide and widespread integration of digital health promotion, concerns about privacy, ageism, and safety should be addressed fully.

Through this paper, it is hoped that we can provide a deeper understanding of the countries’ policies and how we can learn from each other in the evolution of the digital health world.

### 5.3. Strength and Limitation

Our study enables readers to develop a broad understanding of the development and current status of Japan, Korea, Singapore, and Thailand in terms of their digitalization and health promotion journey. In the fast digitizing and aging world, the paths of these countries that were mapped can be a great source for countries in a similar context to adopt and learn from. However, with the nature of scoping review, our research reviewed the broad topic, which is the overall digitization and health promotion journey. In the future, a review of digital empowerment and digital health promotion policies that targeted older people might be necessary in terms of building a digitally inclusive healthy aging society.

### 5.4. Implication for Policy and Practice

Governments are promoting the use of digital technology in attempts to secure health, social welfare, lifelong education, and continuing employment for the older people. Getting older people online has never been more urgent. People themselves are real keys to making digital transformation come true. Therefore, empowerment of older people is essential to minimize digital gaps and policy gaps. Our review highlighted that digitization is a compulsory part of healthy ageing policies and programs. Further research is required to identify integration mechanisms. The policy paths and their impact that we reported in this study may share followable lessons and shed a light on social policies of several countries towards healthy aging communities by closing the gray digital gap.

## 6. Conclusions

Empowering older persons to take control over their health and do what they value doing is indivisible with the digital access, skill, and confidence in new digital era. National policy is the beginning of this empowerment. Japan, Korea, Singapore, and Thailand, though they have different country contexts in terms of economic, culture, and other aspects, are all experiencing the shift of demographic change and continuous digitalization. We hope that the paths that we identified and the journey summarized in this paper would be great sources to learn from and share, leading to the digitally inclusive, healthy aging communities (DIHAC) in Asia and globally.

## Figures and Tables

**Table 1 ijerph-19-16995-t001:** Main review sources for each country.

	ICT Policy	Health Promotion Policy
Japan	Ministry of Internal Affairs and Communications	Ministry of Health, Labour and Welfare; National Institute of Health and Nutrition
Korea	Ministry of Science and ICT	Ministry of Health and Welfare; Korea Institute for Health and Social Affairs
Singapore	Ministry of Communication and Information; Infocomm Media Development Authority	Ministry of Health; Health Promotion Board
Thailand	Ministry of Digital Economy and Society; Digital Government Development Agency	Ministry of Public Health
Other sources	International Telecommunication Union (ITU); United Nation E-Government Knowledgebase	World Health Organization (WHO); United Nations Department of Economic and Social Affairs

**Table 2 ijerph-19-16995-t002:** Current national ICT policy of Japan, Korea, Singapore, and Thailand.

	Japan	Singapore	Korea	Thailand
Title	Priority Policy Program for Realizing Digital Society	Smart Nation Strategy	Digital Government master plan	Digital Economy and Society Development Plan
Time	June 2021	November 2018	2021–2025	March 2018
Vision	A society where each citizen can choose services that satisfy his/her demands and achieve various happiness through digital technology	Build a smart nation	Digital, the door to a better world	Transform towards a digital Thailand
Pillars/Strategy	(1) Digitalization of Public Services for Citizens. (2) Digitalization of Lifestyle. (3) Digitalization of Industries. (4) System and Technology for the Digital Society. (5) Lifestyle and Human Resources.	(1) Digital Economy (Digital Economy Framework for Action). (2) Digital government (Digital Government Blueprint). (3) Digital Society (Digital Readiness Blueprint).	(1) Implementing intelligent public services. (2) Facilitating databased government. (3) Strengthening the foundation of digital transformation	(1) Build country-wide high-capacity digital infrastructure. (2) Boost the economy with digital technology. (3) Create a quality and equitable society through digital technology. (4) Transform into digital government. (5) Develop workforce for the digital era. (6) Build trust and confidence in the use of digital technology.

**Table 3 ijerph-19-16995-t003:** Current national health promotion policies of Japan, Korea, Singapore, and Thailand.

	Japan	Korea	Singapore	Thailand
Sector	The Ministry of Health, Labor and Welfare	The Ministry of Health and Welfare	Ministry of Health	The Ministry of Public Health
Title	Health Japan 21 (the second term)	Health Plan 2030	Healthier SG	Twenty-year National Strategic Plan for Public Health
Time	2013–2023	2021–2030	Announced on March 2022	2017–2036
Vision	Realization of a society with vitality in which every person supports each other and can lead a healthy and blissful life	A society where all people enjoy lifelong health	Making Singaporeans healthier	A key health agency that mobilizes public and social engagement for the health and well-being of Thai people
General goals	(1) Extension of healthy life expectancy. (2) Reduction of health disparities.	(1) Extension of healthy life expectancy. (2) Improvement in health equity (gap in income and healthy life expectancy between regions).	Promoting overall healthier living while targeting specific sub-populations.	Healthier People. Happier Health Care Workers. Sustainable Health System.

## Data Availability

DIHAC study related data, progress, publications, and other research related information can be accessed via DIHAC study website (https://digital-ageing.com/ (accessed on 10 November 2022)) [16].

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
