# Peer review of "Digital-Based Policy and Health Promotion Policy in Japan, the Republic of Korea, Singapore, and Thailand: A Scoping Review of Policy Paths to Healthy Aging"

_ijerph, 2022, doi:10.3390/ijerph192416995_

Round 1

Reviewer 1 Report

The manuscript ‘Digital-based policy and health promotion policy in Japan, the Republic of Korea, Singapore, and Thailand: A scoping review of policy paths to Healthy Ageing’. This manuscript is significant in the public health context. I have the following suggestions.

Define 2W1H in the abstract.

Introduction: Well written. Abbreviate ICT

Methods: The methods section needs to improve

What are your search strategies?  

Which are the databases and grey literature you search for your articles?  

Mention the study inclusion and exclusion criteria.

How did you screen the articles?

Results

Provide a Prisma-P check list.

Tricco, AC, Lillie, E, Zarin, W, O'Brien, KK, Colquhoun, H, Levac, D, Moher, D, Peters, MD, Horsley, T, Weeks, L, Hempel, S et al. PRISMA extension for scoping reviews (PRISMA-ScR): checklist and explanation. Ann Intern Med. 2018,169(7):467-473. doi:10.7326/M18-0850.

Results and Discussion:

3. National Digitization Journey, mention it as a subheading under the section Results and Discussion  

The findings are well organised and discussed, and compared with other studies.  

Mention one paragraph on Implications for Policy and Practice

Conclusion

Need to be compact. The remaining write-up moves to the Implication for policy and practice section.

What are the strength and limitations of the studies? 

Author Response

Dear Reviewer:

We would like to express our gratitude to you for taking the necessary time and effort to review the manuscript. We sincerely appreciate all your valuable comments and suggestions, which helped us in improving the quality of the manuscript. 

We addressed every comment accordingly and made the necessary changes in the manuscript. 

Thank you again for your time and effort. 

Reviewer 2 Report

What is 2W1H? Please try to be very accurate about the abbreviations that you use. Please try to include all abbreviations as key words

Abstract

The research gap could be better highlighted in relation to the international literature.

The last paragraph of the introduction should contain a brief description of its next sections

Method

Please try to detail more this section and please try to cite more references regarding the method that you use

The relationship between mandatory COVID-19 vaccination and public health misinformation as regards digital-based and health promotion policies has not been covered, and thus such sources can be cited: Lăzăroiu, G., Mihăilă, R., and BraniÈ™te, L. (2021). “The Language of Misinformation Literacy: COVID-19 Vaccine Hesitancy Attitudes, Behaviors, and Perceptions,” Linguistic and Philosophical Investigations 20: 85–94. doi: 10.22381/LPI2020217. Bratu, S., and Sabău, R. I. (2021). “Is Mandatory COVID-19 Vaccination Socially Acceptable and Ethically Justifiable? Attitudes to and Adoption of Public Health Measures,” Analysis and Metaphysics 20: 187–201. doi: 10.22381/am20202113. Lăzăroiu, G., Mihăilă, R., and BraniÈ™te, L. (2021). “The Language of COVID-19 Vaccine Hesitancy and Public Health Misinformation: Distrust, Unwillingness, and Uncertainty,” Review of Contemporary Philosophy 20: 117–127. doi: 10.22381/RCP2020217.   The approach of the paper is well documented. I would recommend the authors to try to pinpoint in more detail the relevance and importance of these countries for the international market - global market and how they can develop - contribute in increasing the global flows (merchandise, information, capital, human resources, raw materials etc.). I think that the paper is missing a section of discussions and future research agenda, where you could pinpoint some topics worth of investigation in future research from a conceptual, but also from a theoretical perspective. Conclusions How does the paper add value to the literature? How could knowledge be further enhanced based on your paper? Please try to also pinpoint some more limitations and future prospects. Managerial relevance of the paper should also be considered. 

Author Response

Dear reviewer,

We greatly appreciate you for taking the necessary time and effort to review the manuscript. We are sincerely thankful for all your valuable comments and suggestions, which helped us in improving the quality of the manuscript.

We responded to all of your comments and made the necessary change in our manuscript accordingly.

Thank you sincerely again for your time and effort. 

Round 2

Reviewer 1 Report

Thank you for addressing all the suggestions.